# 3D segmentation of perivascular spaces on T1-weighted 3 Tesla MR images with a convolutional autoencoder and a U-shaped neural network

**Marc Joliot**[1,2]                                    MARC.JOLIOT@U-BORDEAUX.FR
[1] *Ginesislab, Bordeaux, France.*
[2] *GIN, IMN, UMR 5293, Univ. Bordeaux, CNRS, CEA, France.*

**Ami Tsuchida**[2]
**Alexandre Laurent**[2]
**Violaine Verrecchia**[1,2]
**Christophe Tzourio**[3,4]
[3] *Bordeaux Population Health, U1219, INSERM, France.*
[4] *Centre Hospitalier Universitaire de Bordeaux, France.*

**Bernard Mazoyer**[1,2,4]                               BERNARD.MAZOYER@U-BORDEAUX.FR
**Philippe Boutinaud**[1,5]                              PBOUTINAUD@FEALINX.COM
[5] *Fealinx, Lyon, France.*

**Editors:** Under Review for MIDL 2021

## Abstract

For its involvement in cognitive deterioration and dementia, assessment of enlarged perivascular spaces (PVSs) has become a major area of interest. We implemented a deep learning model for the 3D segmentation of PVSs in deep white matter. It was trained and tested using T1-weighted magnetic resonance imaging data from 1,832 young adults. The model was trained first based on a CNN autoencoder with the full dataset then with a U-net like architecture trained with a subset of 40 T1-weighted MRI manually annotated images. The Dice coefficient (from a separate test subset of 10 images) was 0.64 for cluster detection. Dice values above 0.90 were reached for detecting PVSs larger than 10 mm$^3$. Using the full dataset, the predicted PVS load showed a high degree of agreement with a semi-quantitative visual rating. Finally, we demonstrated the interoperability of this model using a second dataset.

**Keywords:** perivascular space, deep learning, T1-weighted

## 1. Introduction

Our goal was develop an interoperable, unsupervised, and validated deep learning (DL) model for the detection and quantification of PVSs (Kubie, 1927) in the entire brain white matter volume using the most commonly acquired T1-weighted brain MRI images. The implemented DL model is based on fully convolutional autoencoders and U-shaped networks. The model performance was assessed 1) using a sparse (from 20 to 40) number of manually traced data for the training 2) for increasing VRS sizes. The VRS load prediction computed on 1,782 individuals was compared 1) to a visual rating given by a trained rater and 2) to VRS load predicted on another dataset.

## 2. Methods

Brain T1-weighted images acquired on a 3-Tesla MRI scanner using a three-dimensional MPRAGE sequence MRI data were taken from the MRi-Share database (Tsuchida et al., 2020). A trained investigator performed a voxelwise manual delineation of each deep white matter (DWM) PVS on the raw T1w images of each of 50 individuals. Another investigator visually rated on a 4-level severity scores the global PVS burden for each of the 1,832 individuals of the sample (Zhu et al., 2011).

The DL segmentation model used is a U-shaped convolutional neural network similar to U-net. Encoding and decoding blocks are composed by two 3D convolutions with a 3x3x3 kernel, batch normalization, and a swish activation then a maxpooling layer and a dropout layer (0.1). The full 3D images (batch size 4) are used as inputs, the first encoding block generates 8 images before maxpooling then 16, 32... for a depth of 7 blocks then the decoding blocks use upsampling and convolutional blocks to get back to the original images size (segmentation masks) with cropping and resizing to match the shape of inputs coming from the skip connections from the encoding blocks. An autoencoder with the same architecture (w/o the skip connections) was used to pre-train the layers of the model with non-annotated images, to speed up and to facilitate convergence of the training. Dice loss was used with the Adam optimizer. To prevent overfitting, we used augmentations (flip around the symmetry axis of the brain and small translations) and used dropout in the model architecture. Using a 10 participant's images fully traced testing set we computed the Sorensen-Dice coefficient for 9 thresholds of the prediction map ($Pthr = 0.1$ to $0.9$ in step of 0.1) for both voxel and cluster levels and by thresholding on the cluster size from 0 to $15\text{mm}^3$.

## 3. Results

**Training set size effect.** Training the model was initially with only 10 available annotated subjects (using 5 folds CV) and had difficulty to converge without the autoencoder pre-training, stability of results was attained with the availability of 20 subjects (dice 0.48), then increased with 30 subjects to dice 0.5. **Performance tuning.** Using 40-subject training prediction algorithm, Figure 1.A shows the dice value for each threshold P-thr while filtering the data on the cluster size. Without filtering, the maximum dice was measured at 0.51 / 0.54 (voxel / cluster metrics, respectively, for P-thr at 0.6) and the cluster indexes increased to 0.9 for detecting PVS larger than $10\text{mm}^3$. **Prediction compared to visual rating.** The logistic regression between the number of DWM PVS clusters and the visual grading rating was highly significant ($R_{0.5}^2 = 0.45, p < 0.001$, N=1,782, see Figure 1.B). **Assessment of the prediction database interoperability.** Prediction distribution of the 1,782 subjects (Figure 1.C) was not different from the distribution of 354 subjects of another age- and sex-matched database (Mazoyer et al., 2016) when clusters below $5\text{mm}^3$ where filtered out (Kolmogorov-Smirnov test, d = 0.077, p-value = 0.058).

## 4. Discussion and Conclusion

The segmentation model used in our work were based on the U-net architecture described in (Ronneberger et al., 2015) adapted for 3D images, the main parameters of the model

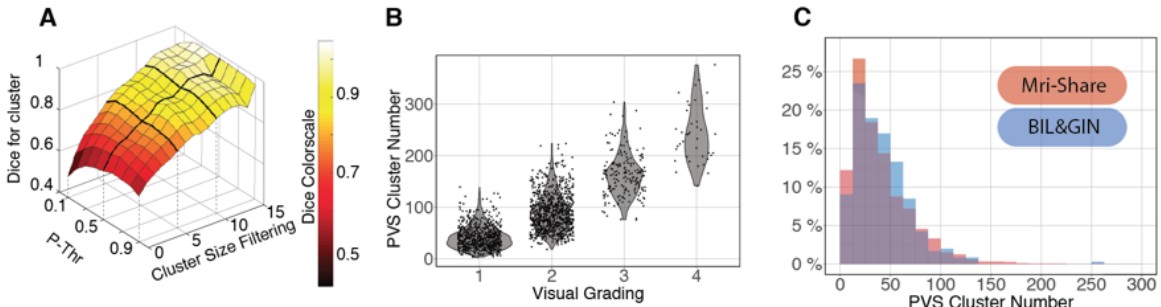

Figure 1: A) Dice versus prediction-threshold and cluster size (10 subjects), B) Logistic regression with visual grading (1782 subjects), C) Interoperability with BIL&GIN database.

(depth, width and number of convolutions) were tuned to fit the constraints imposed by the GPU RAM. The dataset used for training was very limited since the subjects were from a dataset of young adults with very few PVS per subject. Pre-training the model through an autoencoder with the same architecture as the segmentation model was very helpful to ensure a stable convergence of the training with the limited set of annotated images.

To conclude, we implemented and validated an interoperable predictive model for the quantification of PVS using T1-weighted MRI images only that could be used both for routine clinical analysis and for mega- or meta-analysis across datasets.

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
