# OpenReview forum: "3D segmentation of perivascular spaces on T1-weighted 3 Tesla MR images with convolutional autoencoder and U-shaped neural networks"
_MIDL.io/2021/Conference/Short — Submitted to MIDL 2021_

### Official Review · Reviewer_bxHB · 2021-04-22

**Confidence:** 3
**Final Rating:** 2

**Summary:**

The authors present a method for segmentation of perivascular spaces in deep white matter using a U-Net, pre-trained by auto-encoding a large MRI dataset. They compare the performance of the method using different training set sizes and correlate the network results to a visual rating for each subject, as determined in a previous study. They also compare the distribution of the results to the distribution of results from a different dataset to assess the generalizability of the method.

**Strengths:**

The work shows a strong correlation between the output of the presented method and a visual grade developed in earlier work. Figure 1b presents this correlation in a very clear and convincing way.

The statistical comparison to the results from the BIL&GIN set show that the method generalizes without introducing any major biases to the results.

**Weaknesses:**

The paper is not very well written. Not all abbreviations are defined (e.g. VRS) and many sentences are not correct English. This makes the paper somewhat hard to read.

After "performance tuning", it is not clear what results are actually used for Figure 1b. Is the cluster size filtering experiment just to show the performance is better for larger clusters, or are clusters smaller than 10mm^3 actually discarded? If so, is this because there are too many mistakes in the smaller clusters, or is there also a clinical justification for ignoring smaller clusters?

The experiment set-up for the first experiment (exploring the effect of training set size) is not clear to me. The first sentence in Section 3 states that the model trained with 10 images had difficulty converging without the autoencoder pre-training, and hence no result is given. Was the autoencoder pre-training used for training with 20, 30 and 40 images? If not, how much did the autoencoder pre-training help in the final model? These results seem to be missing.


**Deanonymize Review:**

no

**Justification Of The Rating:**

While the work presents a promising method that is validated on a very large dataset, the paper itself is not very well written. The description of some of the experiments is confusing and it is unclear how much the autoencoder pre-training helps (if at all).

**Paper Type:**

validation/application paper

**Special Issue:**

no

---

### Official Review · Reviewer_pdqZ · 2021-04-29

**Confidence:** 3
**Final Rating:** 2

**Summary:**

In this work, the authors present a deep learning based method for segmenting perivascular space from T1 brain MRI images. Thus, they evaluate a U-net based model with unsupervised pretraining and test it on a labeled dataset. For evaluation, they assess pixel-based metrics as well as cluster-based ones.

**Strengths:**

The authors provide a lot of different evaluation concepts beyond classical dice coefficients, including a correlation between the number of identified clusters and the severity rating of a human expert as well as a cross-dataset comparison.
The figures are nice and the language is good.

**Weaknesses:**

Unfortunately, the manuscript is missing a consistent red line. A lot of different results are reported without motivating them sufficiently or presenting an adequate background. At the same time, core elements of the evaluation remain unclear. How was a cluster defined? Did a higher threshold on a cluster reduces the number of clusters? If yes, what is a sufficient threshold for the clusters from a clinical point of view? As long as this remains unclear, an interpretation of the Dice score is barely possible.
Additionally, I am confused about the size of the dataset. In 2, it is stated that the threshold level analysis spans 10 individuals. However, in 3. it is claimed that Fig. 1A was computed for 40 individuals. Furthermore, it is stated in 3 that the convergence was difficult for 10 individuals while the accuracy improves with a higher number, which is quite obvious. Why are the results still claimed to be computed on 10 individuals? And if the model converged for more individuals without pretraining, why is the pretraining still proposed?

Another issue is the insufficient embedding into previous work. I know that there is not much space, but at least the following paper of Lian et al. should have been addressed:

https://www.ncbi.nlm.nih.gov/pmc/articles/PMC6430123/

Lian et al. also used a U-Net like architecture for PVS segmentation with higher dice scores. Why is this the case?
Furthermore, I am wondering why the authors did not utilize No New-Net (https://arxiv.org/abs/1809.10483) as the current state-of-the-art for 3D medical image segmentation.

**Deanonymize Review:**

yes

**Detailed Comments:**

In general, the paper features a clear language. There are relevant details missing, please see above under "Weaknesses".

**Justification Of The Rating:**

Even though the effort of the authors providing a profound evaluation is definitely worth mentioning, I cannot vote for accepting this manuscript in its current form. It misses a red line, the result section is overcrowded and thus confusing while core details of the method as well as the clinical embedding are missing. Hence, it is not clear how the results were computed and how to interpret them. Therefor, I vote for rejecting this work.
However, I want to encourage the authors to keep up with their work! It is visible that a lot of work and effort was taken conducting this study. This is definitely worth a publication, but not in the form of this manuscript.

**Paper Type:**

validation/application paper

**Special Issue:**

no

---

### Meta-Review · Program_Chairs · 2021-05-09

**Recommendation:** Reject
**Confidence:** 4

**Metareview:**

There is not sufficient enthusiasm from the reviewers for this paper, both reviewers recommended rejection.

---

### Decision · Program_Chairs · 2021-05-11

Reject